# Anticancer Effect by Combined Treatment of *Artemisia annua* L. Polyphenols and Docetaxel in DU145 Prostate Cancer Cells and HCT116 Colorectal Cancer Cells

Eun Joo Jung [1], Hye Jung Kim [2], Sung Chul Shin [3], Gon Sup Kim [4], Jin-Myung Jung [5], Soon Chan Hong [6], Ky Hyun Chung [7], Choong Won Kim [8,†] and Won Sup Lee [1,*]

[1] Department of Internal Medicine, Institute of Medical Science, Gyeongsang National University Hospital, Gyeongsang National University College of Medicine, 15 Jinju-daero 816 Beon-gil, Jinju 52727, Republic of Korea; eunjoojung@gnu.ac.kr

[2] Department of Pharmacology, Institute of Medical Science, Gyeongsang National University College of Medicine, Jinju 52727, Republic of Korea

[3] Department of Chemistry, Research Institute of Life Science, Gyeongsang National University, Jinju 52828, Republic of Korea; sshin@gnu.ac.kr

[4] Research Institute of Life Science, College of Veterinary Medicine, Gyeongsang National University, Jinju 52828, Republic of Korea; gonskim@gnu.ac.kr

[5] Department of Neurosurgery, Institute of Medical Science, Gyeongsang National University Hospital, Gyeongsang National University College of Medicine, Jinju 52727, Republic of Korea; gnuhjjm@gnu.ac.kr

[6] Department of Surgery, Institute of Medical Science, Gyeongsang National University Hospital, Gyeongsang National University College of Medicine, Jinju 52727, Republic of Korea; hongsc@gnu.ac.kr

[7] Department of Urology, Institute of Medical Science, Gyeongsang National University Hospital, Gyeongsang National University College of Medicine, Jinju 52727, Republic of Korea; kychung@gnu.ac.kr

[8] Department of Biochemistry, Institute of Medical Science, Gyeongsang National University College of Medicine, Jinju 52727, Republic of Korea

\* Correspondence: lwshmo@gnu.ac.kr; Tel.: +82-55-750-8733

† Passed away on May 2022.

**Abstract:** Docetaxel (DTX), a semi-synthetic analogue of paclitaxel (taxol), is known to exert potent anticancer activity in various cancer cells by suppressing normal microtubule dynamics. In this study, we examined how the anticancer effect of DTX is regulated by polyphenols extracted from Korean *Artemisia annua* L. (pKAL) in DU145 prostate cancer cells (mutant p53) and HCT116 colorectal cancer cells (wild-type p53). Here, we show that the anticancer effect of DTX was enhanced more significantly by pKAL in HCT116 cells than in DU145 cells via phase-contrast microscopy, CCK-8 assay, Western blot, and flow cytometric analysis of annexin V/propidium iodide-stained cells. Notably, mutant p53 was slightly downregulated by single treatment of pKAL or DTX in DU145 cells, whereas wild-type p53 was significantly upregulated by pKAL or DTX in HCT116 cells. Moreover, the enhanced anticancer effect of DTX by pKAL in HCT116 cells was significantly associated with the suppression of DTX-induced p53 upregulation, increase of DTX-induced phospho-p38, and decrease of DTX-regulated cyclin A, cyclin B1, AKT, caspase-8, PARP1, GM130, NF-κB p65, and LDHA, leading to the increased apoptotic cell death and plasma membrane permeability. Our results suggest that pKAL could effectively improve the anticancer effect of DTX-containing chemotherapy used to treat various cancers expressing wild-type p53.

**Keywords:** *Artemisia annua* L. polyphenols; docetaxel; anticancer effect; p53; prostate cancer; colorectal cancer; chemotherapy

## 1. Introduction

Paclitaxel (PTX), a phytochemical extracted from the bark of the Pacific yew tree, belongs to microtubule-stabilizing agents that promote the polymerization of tubulin and prevent microtubule depolymerization, leading to the suppression of microtubule

dynamics [1–3]. Docetaxel (DTX), a semi-synthetic analogue of PTX, also belongs to microtubule-stabilizing agents and is widely used as a chemotherapy drug to suppress various types of cancer, including castration-resistant prostate cancer, breast cancer, head and neck cancer, stomach cancer, and non-small-cell lung cancer [4–9]. Like PTX, DTX suppresses microtubule dynamics, induces cell cycle arrest in the G2-M phase, inhibits cell division, and induces apoptotic cell death [3,10]. DTX-induced apoptotic cell death is related to the modulation of p53; DTX upregulates p53 in human KB epidermoid and HCC1937 breast cancer cells expressing wild-type p53 and downregulates p53 in HT-29 colon cancer cells expressing mutant p53 [11]. It is known that the anticancer effect of DTX is enhanced by flavopiridol, a cyclin-dependent kinase inhibitor, which is associated with a rapid decrease of cyclin B1/cdc-2 kinase activity [12]. However, cyclin B1 is increased by DTX in prostate cancer cells, and this phenomenon is implicated in DTX-induced apoptosis [13]. In addition, DTX upregulates acetylated α-tubulin (Ac-α-Tubulin), a maker of microtubule stabilization, and the anticancer effect of DTX is promoted by the combined treatment with trichostatin A (TSA), a histone deacetylase inhibitor, through upregulation of Ac-α-Tubulin in A549 lung adenocarcinoma cells [14–16]. However, tubulin acetylation upregulated by TSA is also known to be associated with apoptosis resistance to PTX in H460 and A549 lung cancer cells by stabilizing Mcl-1 [17]. Therefore, the anticancer mechanism for DTX may depend on the regulation of cell fate determinants in different types of cancer cells and culture conditions.

Polyphenols, plant-derived natural products, have shown anticancer effects in various cancer cells and have potential activity as a source of less cytotoxic chemotherapy drugs [18–21]. Intake of foods rich in polyphenols, such as fruit, tea, and coffee, has been known to be associated with the reduction of the cancer incidence, cardiovascular disease, and neurodegeneration [22–25]. Polyphenolic compounds can be simply classified into flavonoids and non-flavonoids, and can be subdivided into various subclasses depending on their chemical structure and number of phenol groups [26,27]. It is known that synergistic anticancer effect is induced by combination treatment of PTX with flavonoid-type polyphenols such as flavone, flavonol, baicalein, luteolin, kaempferol, and quercetin [28,29]; in addition, the anticancer effect of DTX is synergistically induced by combined treatment with polyphenols such as baicalein, resveratrol, and quercetin in pancreatic cancer cells and prostate cancer cells [30–32]. However, studies on the efficient anticancer effect and mechanism by the combined treatment of DTX and various types of polyphenols are still needed to further improve DTX-containing chemotherapy used in the treatment of various cancers.

*Artemisia annua* L., an annual herb, has been traditionally used for the prevention and treatment of infectious diseases, and has exhibited various biological activities including anti-malarial, anti-tumor, anti-microbial, and immunomodulatory properties [33,34]. We have previously reported that the anticancer effect of polyphenols extracted from Korean *Artemisia annua* L. (pKAL) was higher in HCT116 colorectal cancer cells expressing wild-type p53 than null-type p53 by promoting the cleavage of PARP1 and lamin A/C [35]. In this study, we investigated the anticancer effect by the combined treatment of DTX and pKAL in DU145 prostate cancer cells (mutant p53) and HCT116 colorectal cancer cells (wild-type p53). As a result, the anticancer effect of DTX was more effectively enhanced in HCT116 cells than in DU145 cells by less toxic pKAL in a concentration-dependent manner. Moreover, the enhanced anticancer effect by the combined treatment of DTX and pKAL in HCT116 cells was related to the suppression of DTX-induced p53 upregulation, promotion of apoptotic cell death signaling through decrease of AKT, caspase-8, PARP1, GM130, NF-κB p65, and LDHA, and inhibition of G2-M cell cycle progression through decrease of cyclin A and cyclin B1. Our results suggest that the combination treatment of DTX and pKAL has the potential to be used to effectively improve DTX-containing chemotherapy for various cancers, especially in the cancer treatment containing wild-type p53 that is activated by pKAL as well as DTX.

## 2. Materials and Methods

### 2.1. Materials

Eagle's minimum essential medium (EMEM, 30-2003), Trypsin-EDTA solution (30-2101), and dimethyl sulfoxide (DMSO) were purchased from the American Type Culture Collection (ATCC, Manassas, VA, USA). RPMI 1640 medium was purchased from HyClone (Logan, UT, USA). Penicillin-Streptomycin (10,000 U/mL) and TrypLE™ Express Enzyme with phenol red were purchased from Thermo Fisher Scientific (Grand Island, NY, USA). Propidium iodide (PI), necrostatin-1, phosphate-buffered saline (PBS), and sodium dodecyl-sulfate (SDS) were purchased from Sigma-Aldrich (St. Louis, MO, USA). Cell counting kit-8 (CCK-8) was purchased from Dojindo (Kumamoto, Japan). Protein assay dye reagent concentrate, 30% acrylamide/bis solution 29:1, 2-mercaptoethanol, and bromophenol blue were purchased from Bio-Rad (Hercules, CA, USA). Tween-20 and DMSO were purchased from Amresco (Solon, OH, USA). An amount of 0.22 µm nitrocellulose (NC) transfer membrane was purchased from GVS Life Sciences (Sanford, ME, USA). Enhanced chemiluminescence (ECL) Ottimo Western blot detection kit was purchased from TransLab (Daejeon, Republic of Korea). X-ray film (CP-BU NEW) was purchased from AGFA (Mortsel, Belgium). Annexin V-Fluos was purchased from Roche (Mannheim, Germany). Dishes, plates, tubes and pipettes for cell culture were purchased from SPL Life Sciences (Pocheon, Republic of Korea) or Thermo Fisher Scientific (Rockford, IL, USA). Ac-α-Tubulin (6-11B-1), cyclin B1 (GNS1), p53 (DO-1), PARP1 (F-2), cyclin A (H-432), cyclin D1 (A-12), Akt1/2/3 (H-136), caspase-8 (8CSP03), NF-κB p65 (F-6), and GAPDH (FL-335) antibodies were purchased from Santa Cruz Biotechnology (Santa Cruz, CA, USA). Phosphorylated JNK on Thr-183 and Tyr-185 residues (Phospho-JNK) antibody was purchased from Bioworld (St. Louis Park, MN, USA). GM130 was purchased from BD Biosciences (San Jose, CA, USA). Phospho-p38, p38, and LDHA were purchased from Cell Signaling Technology (Beverly, MA, USA). Secondary goat anti-rabbit and anti-mouse horseradish peroxidase (HRP) conjugates were purchased from Bio-Rad (Hercules, CA, USA).

### 2.2. pKAL Compounds

*Artemisia annua* L. polyphenols (pKAL) were extracted from mixed tissues including roots, stems, leaves, and flowers of *Artemisa annua* L. (Gaddongsook) grown in Jinju, Korea by Prof. Sung Chul Shin, as previously reported [36]. Briefly, the mixed tissues were lyophilized, ground, and extracted with 70% methanol for 20 h at 60 °C. The extract was filtered through a glass funnel and concentrated at 35 °C using a rotary evaporator. Fat components were removed by extraction three times with equal amounts of *n*-hexane and methylene chloride. After extracting three times with ethyl acetate, it was dried with anhydrous magnesium sulfate and identified by liquid chromatography-tandem mass spectrometry (LC/MS/MS). As a result, pKAL compounds isolated from Gaddongsook are as follows: Caffeic acid, Quercetin-3-O-galactoside, Mearnsetin-glucoside, Kaempferol-3-O-glucoside, Ferulic acid, Isorhamnetin-glucoside, Diosmetin-7-O-glucoside, Luteolin-7-O-glucoside, Quercetin, Quercetagetin-3-O-methyl ether, Luteolin, 8-methoxy-kaempferol, Quercetagetin-5,3-di-O-methyl ether, Kaempferol, 3,5-dihydroxy-6,7,4'-trimethoxyflavone, 3,5-dihydroxy-6,7,3',4'-tetramethoxyflavone, and Isorhamnetin. Quantitative analysis of pKAL compounds using high-performance liquid chromatography (HPLC) was presented in a previous report [37]. The pKAL compounds were dissolved in DMSO solvent at a concentration of 100 mg/mL and used in this study.

### 2.3. Cell Culture

The DU145 human prostate cancer cell line (HTB-81™) was purchased from the American Type Culture Collection (ATCC, Manassas, VA, USA). DU145 cells were maintained in EMEM medium with 1% penicillin/streptomycin and 10% heat-inactivated fetal bovine serum (FBS, 30-2020, ATCC, Manassas, VA, USA). The HCT116 human colorectal cancer cell line was purchased from Korean Cell Line Bank (KCLB No. 10247). HCT116 cells were maintained in RPMI medium containing 300 mg/L of L-glutamine, 25 mM of 4-(2-hydroxyethyl)piperazine-1-ethane-sulfonic acid (HEPES), 25 mM of sodium bicarbonate

(NaHCO$_3$), 1% penicillin/streptomycin, and 10% heat-inactivated FBS (Thermo Fisher Scientific, Grand Island, NY, USA). DU145 and HCT116 cells were grown on a culture dish in a 37 °C incubator supplemented with 5% CO$_2$ in a humidified atmosphere.

*2.4. Phase-Contrast Light Microscopy*

Cell morphology was analyzed by phase-contrast light microscopy (EVOS XL Core, Thermo Fisher Scientific, Oslo, Norway) in a 10× objective (Inf Plan Achro 10× LWD PH, 0.25 NA/6.9 WD) with 150× amplification.

*2.5. Cell Viability Analysis*

Cells grown on a 24-well dish were incubated with maintenance medium containing 10% CCK-8 reagent for 1.5 h in a 37 °C CO$_2$ incubator. The reaction solution (100 µL each) was then transferred to a 96-well dish and was analyzed by measuring the absorbance at OD$_{485\,nm}$ using a CHAMELEON microplate reader (Hidex).

*2.6. Western Blot and Densitometry Analysis*

Whole cells (attached and floating cells) were extracted with 1× SDS sample buffer (31.25 mM Tris, pH 6.8/5% glycerol/1% SDS/2.5% 2-mercaptoethanol/0.005% bromophenol blue), boiled for 5–10 min at 95 °C, and vortexed well to be homogenized. After quantification using protein assay dye reagent, proteins (30 µg each) were separated by SDS polyacrylamide gel electrophoresis (SDS-PAGE) and transferred to an NC membrane at 30 mA for 13–15 h. Membrane was washed twice with PBST buffer (0.1% Tween-20/PBS) for 1 h at room temperature (RT), blocked in blocking buffer (3% skim milk/PBST) for 30 min, and incubated with primary antibody in blocking buffer at 4 °C overnight. And then, the membrane was washed 3 times for 10 min with PBST and incubated with an HRP-conjugated secondary antibody in blocking buffer for 2 h at RT. After washing with PBST, immunofluorescent proteins were detected on X-ray film using the ECL Western blot detection system. For densitometry analysis, the film was scanned at high resolution into TIFF file format. The image was converted into JPEG file format using Photoshop, the image mode was changed to grayscale, and protein bands were quantified using the ImageJ program (version 1.53 k).

*2.7. Flow Cytometric Analysis of Annexin V/PI-Stained Cells*

Whole cells (floating and attached cells) were collected and incubated with annexin V-Fluos in 10 mM HEPES (pH 7.4)/140 mM NaCl/5 mM CaCl$_2$/1 µg/mL PI/PBS solution for 20 min at RT. Annexin V- and PI-stained cell populations were analyzed by FL1-H green and FL2-H red fluorescent light detectors, respectively, using flow cytometry (FACS Calibur, Becton Dickinson).

*2.8. Statistical Analysis*

Data were expressed as mean ± standard deviation. Statistical significance between control and sample was determined using Student's *t*-test. Values of $p < 0.05$ are considered statistically significant.

**3. Results**

*3.1. Anticancer Effect by Combined Treatment of DTX and pKAL in DU145 Prostate Cancer Cells*

First, we investigated how the anticancer effect of DTX is regulated by combined treatment with pKAL in DU145 cells. Cells were treated for 48 h with 0.1% DMSO control solvent, 25 µg/mL pKAL, 50 µg/mL pKAL, and 100 µg/mL pKAL in the absence or presence of 5 nM DTX, and then morphological changes were analyzed via phase-contrast microscopy. As a result, DU145 cells showed different morphological changes on the concentration by a single treatment of pKAL compared to the DMSO-treated control: at 25 µg/mL pKAL treatment, some cells changed to small round cells; at 50 µg/mL pKAL treatment, most cells showed remarkable structural changes in the nucleus and cytoplasm; at 100 µg/mL pKAL

treatment, most cells turned into small round shapes (Figure 1, upper panels). In addition, the combined treatment of DTX and pKAL showed greater morphological changes than those caused by the combined treatment of DTX and 0.1% DMSO control solvent (Figure 1, lower panels). However, in situations where single treatment of 50 or 100 μg/mL pKAL induced significant morphological changes in cells, no significant morphological changes were observed by the combined treatment of DTX and pKAL (Figure 1, compare upper and lower panels). These results show that high doses of pKAL above 50 μg/mL may have high cytotoxicity by themselves, so the combined treatment of DTX and toxic pKAL cannot effectively enhance the anticancer effect in DU145 cells.

**Figure 1.** Morphological changes by combined treatment of pKAL and DTX in DU145 cells: DU145 cells were grown for 48 h on a 10 cm culture dish with the indicated amounts of drugs. Morphology of whole cells (attached and floating cells) was analyzed by phase-contrast light microscopy.

Based on the results in Figure 1, we investigated how the cytotoxicity of DTX is regulated by the amount of less-toxic pKAL in DU145 cells. Cells were treated with 0.1% DMSO, 10 μg/mL pKAL, 20 μg/mL pKAL, and 30 μg/mL pKAL in the absence or presence of 5 nM DTX for 48 h, and then cell viability was analyzed via a CCK-8 assay. The results showed that cell viability was downregulated by single treatment of 10 μg/mL pKAL (84%), 20 μg/mL pKAL (61%), and 30 μg/mL pKAL (36%) (Figure 2, bar graphs 1–4). In addition, downregulation of cell viability by DTX was somewhat promoted by combined treatment with 10 μg/mL pKAL (30%), 20 μg/mL pKAL (30%), and 30 μg/mL pKAL (27%), compared to the 0.1% DMSO-treated control (41%); however, this phenomenon was not significantly promoted in a pKAL concentration-dependent manner (Figure 2, bar graphs 5–8).

To better understand these results, the protein levels of Ac-α-Tubulin, Phospho-JNK, GM130, cyclin B1, p53, and PARP1 were examined using Western blot and densitometry analysis. As a result, the anticancer activity of pKAL was significantly associated with the upregulation of cyclin B1 and downregulation of p53 in DU145 cells expressing mutant p53 (Figure 3, lanes 1–4). In addition, the anticancer activity of DTX was significantly associated with more than two-fold upregulation of Ac-α-Tubulin, GM130, and cyclin B1 (Figure 3, compare lanes 1, 5). Notably, the downregulation of p53 by DTX was somewhat promoted by pKAL, and the upregulation of GM130 and cyclin B1 by DTX was significantly downregulated by pKAL in a concentration-dependent manner (Figure 3, compare lanes 1 and 5–8). However, there was no significant downregulation of cell survival and death regulators such as GM130, cyclin B1, and PARP1 by the combined treatment of DTX and pKAL compared to the 0.1% DMSO-treated control (Figure 3, compare lanes 1 and 5–8). These results show that the anticancer activity by combined treatment of DTX and pKAL may not be efficiently enhanced in DU145 cells expressing mutant p53 due to no significant alteration of cell survival and death determinants compared to DMSO-treated control.

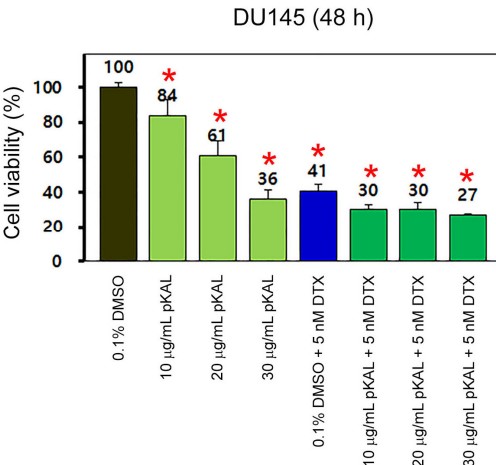

**Figure 2.** Regulation of cell viability by combined treatment of pKAL and DTX in DU145 cells: DU145 cells were grown for 48 h on a 24-well plate with the indicated amounts of drugs. Cell viability was analyzed using a CCK-8 assay in triplicate tests. Statistical significance between control and sample was determined using Student's *t*-test, * $p < 0.05$.

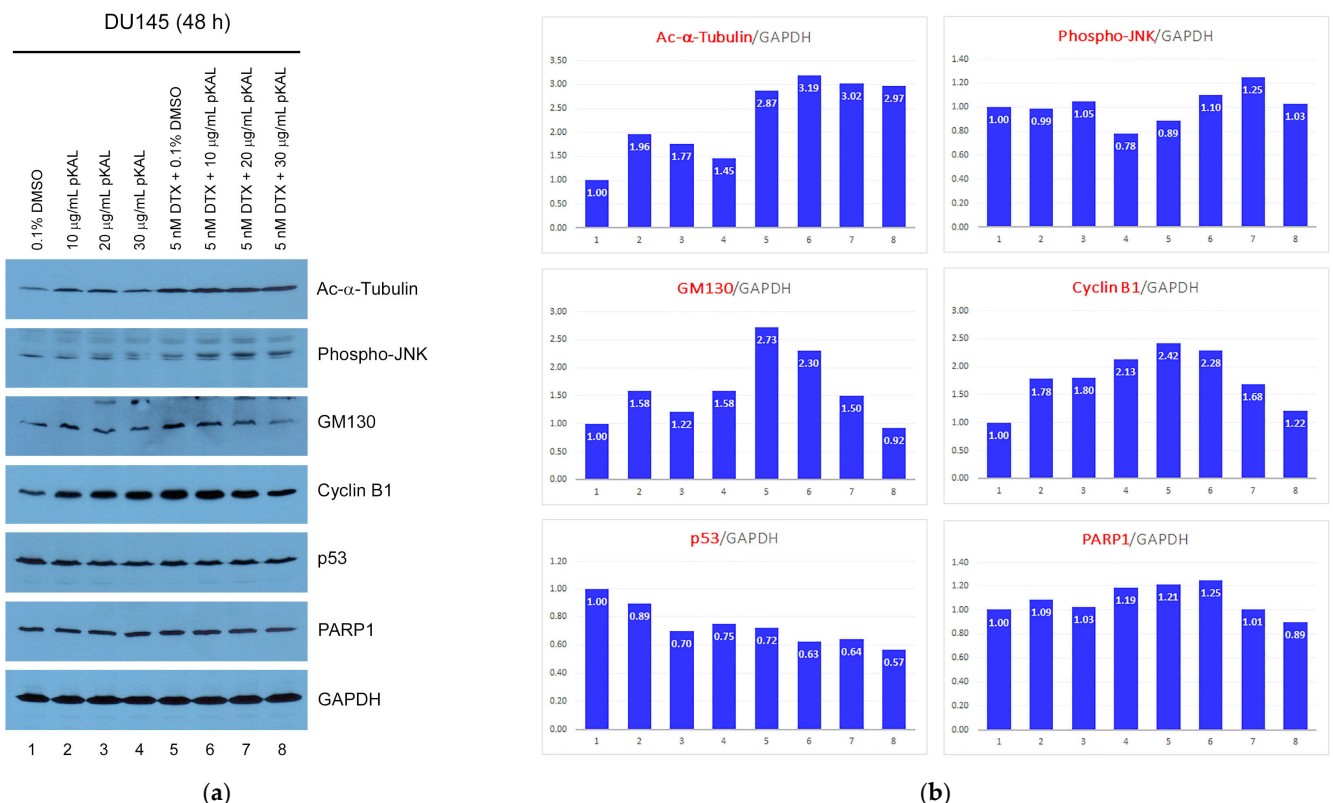

(**a**)           (**b**)

**Figure 3.** Regulation of protein levels by combined treatment of pKAL and DTX in DU145 cells: (**a**) DU145 cells were grown for 48 h on a 10 cm culture dish with the indicated amounts of drugs. Whole cell extracts were prepared using 1× SDS sample buffer and analyzed via Western blot using the indicated antibodies; (**b**) densitometry analysis of protein bands in the panels of (**a**) using the ImageJ program (version 1.53 k).

### 3.2. Anticancer Effect by Combined Treatment of DTX and pKAL in HCT116 Colorectal Cancer Cells

Next, we examined how the anticancer effect of DTX is regulated by the amount of less-toxic pKAL in HCT116 cells expressing wild-type p53. Cells were treated with 0.1% DMSO, 10 µg/mL pKAL, 20 µg/mL pKAL, and 30 µg/mL pKAL in the absence or presence of 3 nM

DTX for 72 h, and then morphological changes were analyzed via phase-contrast microscopy. As a result, compared to the DMSO-treated control, cell morphology was somewhat changed by single treatment of pKAL in a concentration-dependent manner (Figure 4, upper panels). In addition, compared to the combined treatment of DTX and DMSO, cell morphology was significantly altered by the combined treatment of DTX and pKAL in a pKAL concentration-dependent manner (Figure 4, lower panels). Moreover, there were significant morphological differences between single treatment with pKAL and combination treatment with DTX and pKAL; especially, combined treatment with DTX and 30 μg/mL pKAL resulted in a decrease in cell number and remarkable structural changes in the plasma membrane, cytoplasm, and nucleus (Figure 4, compare upper and lower panels). These results show that the less-toxic pKAL enhances the anticancer activity of DTX by inducing abnormal intracellular structural changes in HCT116 cells expressing wild-type p53.

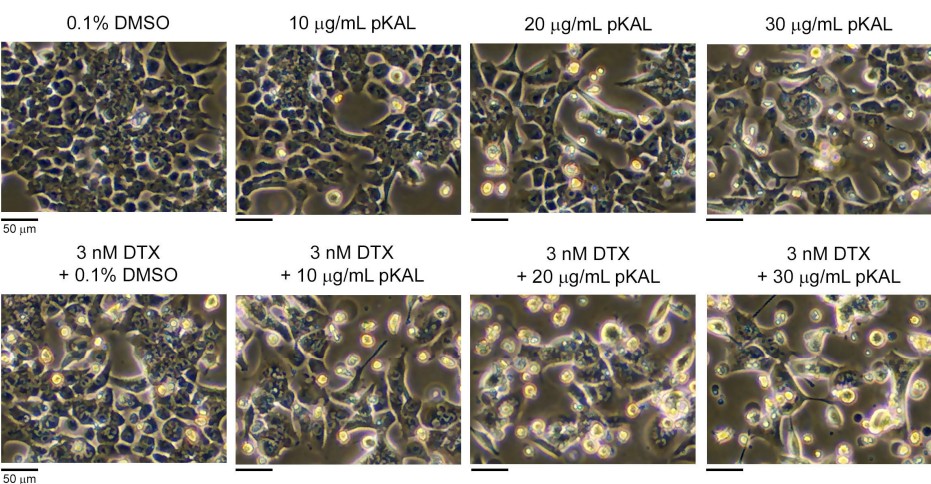

**Figure 4.** Morphological changes by combined treatment of pKAL and DTX in HCT116 cells: HCT116 cells were grown for 72 h on a 10 cm culture dish with the indicated amounts of drugs. Morphology of whole cells (attached and floating cells) was analyzed by phase-contrast light microscopy.

To better understand these results, we performed a Western blot analysis using various antibodies involved in cell survival and death signaling in whole extracts of HCT116 cells expressing wild-type p53, and the protein bands were quantified using the ImageJ program (version 1.53 k). As a result, the anticancer activity of pKAL in HCT116 cells was associated with a more than 2-fold significant upregulation of p53 and phospho-p38, and downregulation of Ac-α-Tubulin and cyclin D1 (Figure 5a,b, lanes 1–4). In addition, the anticancer effect of DTX was associated with significant upregulation of p53 and phospho-p38, and slight upregulation of Ac-α-Tubulin, cyclin A, and cyclin B1 (Figure 5a,b, compare lanes 1, 5). Moreover, the enhanced anticancer activity of DTX by pKAL was significantly associated with an increase in phospho-p38 and Ac-α-Tubulin and a decrease in cyclin A and B1 in a pKAL concentration-dependent manner, compared with DTX single treatment (Figure 5a,b). Notably, unlike cyclin A and cyclin B1, the enhanced anticancer activity of DTX by pKAL was not significantly affected by cyclin D1 (Figure 5a,b).

In addition, the protein levels of AKT, caspase-8, PARP1, GM130, NF-κB p65, and LDHA were downregulated by single treatment of pKAL or DTX, and this phenomenon was more promoted by the combined treatment of DTX and pKAL in a pKAL concentration-dependent manner (Figure 5c,d). Especially, all of the downregulated protein levels by the combined treatment of 3 nM DTX and 30 μg/mL pKAL were significantly lower compared to treatment with 0.1% DMSO control solvent due to the enhanced anticancer activity of DTX by pKAL (Figure 5c,d, compare lanes 1,8). These results show that pKAL enhances the anticancer activity of DTX by inhibiting cell survival signaling and promoting the apoptotic cell death process through the downregulation of AKT, GM130, NF-κB p65, LDHA, caspase-8, and PARP1.

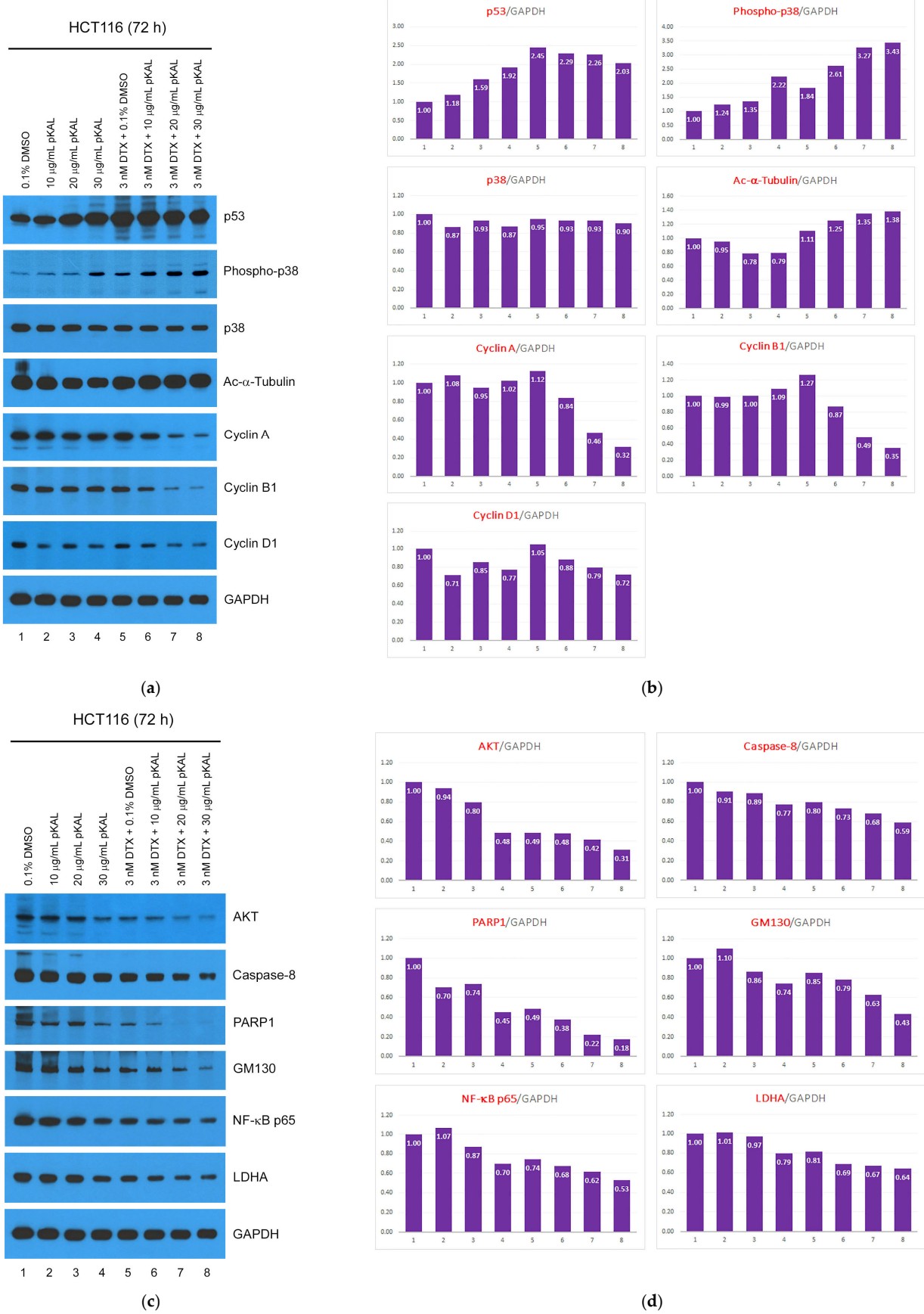

**Figure 5.** Regulation of protein levels by combined treatment of pKAL and DTX in HCT116 cells: (**a**,**c**) HCT116 cells were grown for 72 h on a 10 cm culture dish with the indicated amounts of drugs.

Whole cell extracts were prepared using $1\times$ SDS sample buffer and analyzed via Western blot using the indicated antibodies; (**b**,**d**) densitometry analysis of protein bands in the panels of (**a**,**c**) using the ImageJ program (version 1.53 k).

To better understand the anticancer mechanism enhanced by the combined treatment of DTX and pKAL, HCT116 cells were pretreated with 0.1% DMSO or 10 µM necrostatin-1, an inhibitor of necroptosis, for 16 h, then split and treated again with 0.1% DMSO, 30 µg/mL pKAL, 3 nM DTX, and combination of 30 µg/mL pKAL and 3 nM DTX for 72 h. Cells were co-stained with annexin V and PI, and analyzed using flow cytometry. As shown in Figure 6, in DMSO-pretreated HCT116 cells, annexin V-stained cells were increased by single treatment of pKAL (11.41%) or DTX (13.00%) compared to the DMSO-treated control (4.88%), and this phenomenon was enhanced by the combined treatment of pKAL and DTX (21.86%) (second panels in DMSO-pretreated HCT116). Similarly, in necrostatin-1-pretreated HCT116 cells, annexin V-stained cells were more significantly increased by the combined treatment of pKAL and DTX (23.82%) compared to the single treatment of pKAL (12.44%) or DTX (16.52%) (second panels in necrostatin-1-pretreated HCT116). Moreover, in DMSO-pretreated HCT116 cells, PI-stained cells were remarkably increased by the combined treatment of pKAL and DTX (51.92%) compared to the single treatment of pKAL (28.30%) or DTX (31.49%) (third panels in DMSO-pretreated HCT116). Similarly, in necrostatin-1-pretreated HCT116 cells, PI-stained cells were more significantly increased by the combined treatment of pKAL and DTX (52.68%) compared to the single treatment of pKAL (29.46%) or DTX (34.69%) (third panels in necrostatin-1-pretreated HCT116). However, annexin-V- and PI-stained cell populations were not significantly altered by either single or combination treatment of pKAL and DTX between DMSO and necrostain-1 pretreatment (compare DMSO-pretreated and necrostatin-1-pretreated panels). These results show that higher cell death by the combined treatment of pKAL and DTX than single treatment of pKAL or DTX was significantly associated with apoptosis and increase of plasma membrane permeability, which induced PI uptake into dead or dying cells; however, this phenomenon was not significantly related with necroptosis in HCT116 cells expressing wild-type p53.

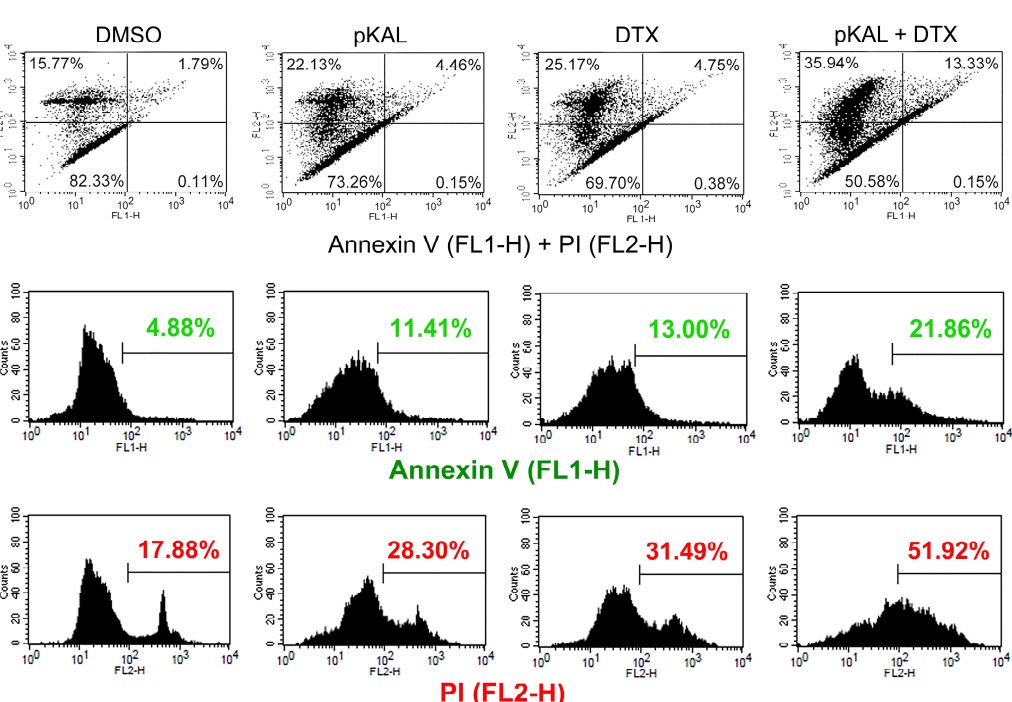

**Figure 6.** *Cont.*

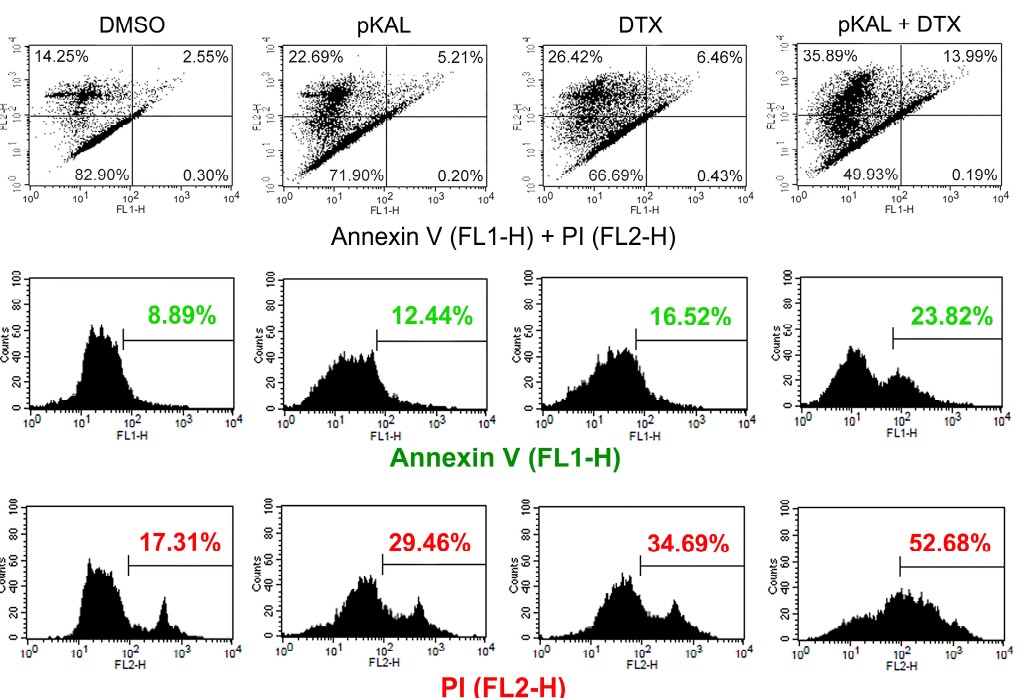

**Figure 6.** Effects of pKAL, DTX, and necrostatin-1 on apoptosis and PI uptake in HCT116 cells. HCT116 cells were grown for 16 h on a 10 cm culture dish with 0.1% DMSO or 10 μM necrostatin-1, then split and grown again for 72 h on a 6-well plate with 0.1% DMSO, 30 μg/mL pKAL, 3 nM DTX, and combination of 30 μg/mL pKAL and 3 nM DTX. Cells were co-stained with annexin V and PI, and analyzed using a flow cytometry: first panels, dot plots for co-stained cells with annexin V (FL1-H) and PI (FL2-H); second panels, histogram of annexin V-stained cells; third panels, histogram of PI-stained cells.

## 4. Discussion

DTX is a chemotherapy agent widely used in the treatment of various malignant cancers, as well as castration-resistant prostate cancer, by inducing microtubule stabilization and G2-M phase cell cycle arrest [5,6]. The anticancer activity of DTX is known to be improved by combined treatment with less-toxic natural polyphenols, such as resveratrol, in prostate cancer cells [38]. Many studies on the biological role of *Artemisia annua* L. were accomplished using artemisinin and its derivatives, which exerted anticancer activity [39,40]. However, little is known about a role of polyphenols isolated from *Artemisia annua* L. on the anticancer effect of DTX. In this study, we investigated how the anticancer effect of DTX is regulated by pKAL in castration-resistant DU145 prostate cancer cells (mutant p53) and HCT116 colorectal cancer cells (wild-type p53). As a result, there were morphological changes and cytotoxicity by single treatment of pKAL or DTX for 48 h in DU145 cells; however, there were no significant synergistic morphological changes and cytotoxicity by the combined treatment of 5 nM of DTX and 10 to 100 μg/mL of pKAL compared to single treatment of pKAL (Figures 1 and 2). In contrast, cell morphology was significantly changed in a pKAL concentration-dependent manner by the combined treatment of 30 μg/mL pKAL and 3 nM DTX for 72 h compared to single treatment of pKAL or DTX in HCT116 cells, resulting in increased apoptotic cell death and plasma membrane permeability (Figures 4 and 6).

It is known that DTX induces p53-dependent apoptosis in human epithelial cancer and prostate cancer cells; in addition, DTX induces p53-independent apoptosis in p53-null PC3 prostate cancer cells [11,41–43]. In this study, our results showed that the anticancer effect of DTX was more effectively enhanced by pKAL in HCT116 colorectal cancer cells (wild-type p53) than in DU145 prostate cancer cells (mutant p53) when wild-type p53 was

upregulated by the single treatment of pKAL as well as DTX in HCT116 cells; however, mutant 53 was slightly downregulated by the single treatment of pKAL or DTX in DU145 cells (compare Figure 3a,b and Figure 5a,b). Notably, the enhanced anticancer activity of DTX by pKAL in HCT116 cells was associated with the suppression of DTX-induced wild-type p53 upregulation and the increase of DTX-induced p38 phosphorylation (Figure 5a,b). It is known that DTX-induced apoptosis is activated by JNK phosphorylation, but not by p38 phosphorylation, in human melanoma cells [44], and the knock-down of p38 protein by small interference RNA (siRNA) sensitizes DTX-induced apoptosis in prostate cancer cells [42]. Therefore, it is currently unclear whether phospho-p38 upregulated by combined treatment of DTX and pKAL is involved in cell survival or death in HCT116 cells (Figure 5a,b).

Microtubules are assembled from heterodimers of α- and β-tubulin, and DTX-induced tubulin acetylation is associated with drug resistance to DTX in prostate cancer cells [45]. In addition, DTX promotes Ac-α-Tubulin upregulation induced by tanespimycin, and tanespimycin-induced Ac-α-Tubulin has been shown to play a dual role in apoptosis and autophagy depending on its protein level in lung cancer cells [46]. Cyclins and cyclin-dependent kinases (CDKs) play an important role in the regulation of cell cycle at specific phase. Cyclin A is known as a marker for tumor proliferation and prognosis in advanced breast cancer, essential for DNA replication at S phase, and also involved in G2-M cell cycle progression [47]. The tumor suppressor p53 is known to block cancer development by suppressing various types of cyclin/CDK complexes such as cyclin D-CDK4/6, cyclin A/E-CDK2, and cyclin B-CDK1 through upregulation of the CDK inhibitor p21 [48].

In this study, our results showed that Ac-α-Tubulin and cyclin B1 were significantly upregulated by 5 nM DTX in DU145 cells and slightly upregulated by 3 nM DTX in HCT116 cells. Notably, cyclin B1, but not Ac-α-Tubulin, was more significantly downregulated by combined treatment of DTX and pKAL in HCT116 cells expressing wild-type p53 than in DU145 cells, leading to a defect in cell cycle progression (compare Figure 3a,b and Figure 5a,b). Moreover, our results showed that cell death determinant such as PARP1 was significantly downregulated by 3 nM DTX in HCT116 cells containing wild-type p53, and this phenomenon was associated with DTX-induced downregulation of cell survival proteins such as AKT and NF-κB p65 in HCT116 cells (Figure 5c,d). Collectively, our results show that KAL enhances the anticancer effect of DTX through downregulation of cell survival, death, and cell cycle regulators such as AKT, GM130, NF-κB p65, LDHA, caspase-8, PARP1, cyclin A, and cyclin B1 in HCT116 cells expressing wild-type p53 that can be activated by DTX.

## 5. Conclusions

In this study, our results showed that wild-type p53 was significantly upregulated by single treatment of less-toxic pKAL or 3 nM DTX for 72 h in HCT116 cells, whereas mutant p53 was slightly downregulated by single treatment of pKAL or 5 nM DTX for 48 h in DU145 cells. Although Ac-α-Tubulin and cyclin B1 were more increased by 5 nM DTX for 48 h in DU145 cells compared to 3 nM DTX for 72 h in HCT116 cells, cell death determinant PARP1 was more significantly downregulated by DTX in HCT116 cells compared to DU145 cells, and this phenomenon was associated with the downregulation of cell survival proteins such as AKT and NF-κB p65. Moreover, pKAL enhanced the anticancer effect of DTX in HCT116 cells through the suppression of DTX-induced p53 upregulation, increase of DTX-induced phospho-p38, and decrease of DTX-regulated cyclin A, cyclin B1, AKT, caspase-8, PARP1, GM130, NF-κB p65, and LDHA, leading to increased apoptotic cell death and plasma membrane permeability. Our study indicates that less-toxic pKAL could effectively improve the anticancer activity of DTX in various cancer cells expressing wild-type p53 that can be activated by DTX as well as pKAL.

**Author Contributions:** Conceptualization, E.J.J., W.S.L., C.W.K. and K.H.C.; methodology, E.J.J., C.W.K., H.J.K. and S.C.S.; software, E.J.J.; validation, E.J.J. and W.S.L.; formal analysis, H.J.K. and G.S.K.; investigation, E.J.J., S.C.S., H.J.K. and G.S.K.; resources, C.W.K., H.J.K. and W.S.L.; data curation, E.J.J.; writing—original draft preparation, E.J.J.; writing—review and editing, W.S.L. and E.J.J.; visualization, E.J.J., J.-M.J., S.C.H. and K.H.C.; supervision, J.-M.J., S.C.H. and K.H.C.; project administration, E.J.J. and W.S.L.; funding acquisition, W.S.L. All authors have read and agreed to the published version of the manuscript.

**Funding:** This research was funded by the Basic Science Research Program through the National Research Foundation of Korea (NRF) funded by the Ministry of Education (2017R1D1A3B05030971).

**Institutional Review Board Statement:** Not applicable.

**Informed Consent Statement:** Not applicable.

**Data Availability Statement:** Data are contained within the article.

**Conflicts of Interest:** The authors declare no conflict of interest.

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
