# Peer review of "Anticancer Effect by Combined Treatment of Artemisia annua L. Polyphenols and Docetaxel in DU145 Prostate Cancer Cells and HCT116 Colorectal Cancer Cells"

_cimb, doi:10.3390/cimb46020105_

Round 1

Reviewer 1 Report

Comments and Suggestions for Authors

The current study is interesting and could be considered for publication, but before that some major changes need to be made. My suggestions are given below:

The paper must not be written in the first person - Avoid using "we did", "we demonstrated" and use a neutral language "it was realized" "it was demonstrated"

How many repetitions were performed for each experiment?

pKAL and DTX each regulate distinct molecular pathways (e.g., pKAL affects cyclin B1 and p53, while DTX affects Ac-α-tubulin, GM130, and cyclin B1), how do these interactions affect the overall anticancer activity when both drugs are taken together? This is especially important because important cell survival and death determinants (GM130, cyclin B1, PARP1) were not significantly downregulated in the combined treatment scenario?

In your study, you mentioned the upregulation and downregulation of various proteins associated with cell survival and death signaling in response to treatments with pKAL and DTX. Could you explain the specific molecular mechanisms underlying how pKAL enhances the anticancer activity of DTX, particularly with regards to the observed changes in protein levels and their impact on cell survival and apoptosis? I think it would add value to the article.

You mention an increase in DTX-induced p38 phosphorylation in HCT116 cells in the presence of pKAL, what is the functional significance of this p38 phosphorylation in the context of DTX-induced apoptosis, and does it contribute to cell survival or death in HCT116 cells?

The upregulation of Ac-α-tubulin and cyclin B1 by DTX is observed in DU145 and HCT116 cells. What are the specific roles of these proteins in DTX-induced apoptosis, and why does cyclin B1 exhibit a more significant downregulation in HCT116 cells when combined with pKAL?

Author Response

"Please see he attachment."

Reviewer 2 Report

Comments and Suggestions for Authors

Jung and colleagues prepared a manuscript on anticancer effect by combined treatment of Artemisia annua L. polyphenols and docetaxel in DU145 prostate cancer cells and HCT116 colorectal cancer cells. As stated in their abstract and introduction, the main question addressed in the study is focused on the anticancer effects of docetaxel, a semi-synthetic analogue of paclitaxel, are regulated by Artemisia annua L. polyphenols in DU145 prostate cancer cells and HCT116 colorectal cancer cells. Here, it seems to me, it is necessary to remove the first sentence, because it must have a literary reference, and the abstract cannot contain them. It is enough to cite this sentence in the text, which is what the authors did.

The authors are well versed in the research topic, and the topic itself is quite original and relevant and is directly related to the subject of the journal. The introduction justifies the relevance of the study, but rather briefly, and it is not entirely clear what the advantage of this study is. I would like the authors to cover this point more seriously in comparison with the literature data.

The methodology as well as discussion used by the authors is understandable and clear, but there is no conclusion section. I would like the conclusion to be more substantiated, expanded and included in a separate section.

I recommended that this manuscript can be accepted for publication with minor revision. Further efforts are required for improving the quality, grammar and coherence of the manuscript.

Author Response

"Please see he attachment."

Reviewer 3 Report

Comments and Suggestions for Authors

In the following manuscript entitled: Anticancer Effect by Combined Treatment of Artemisia annua 2 L. Polyphenols and Docetaxel in DU145 Prostate Cancer Cells 3 and HCT116 Colorectal Cancer Cells, the authors, examined how the anticancer effect of a semi-synthetic analogue of paclitaxel in DU145 prostate cancer cells (mutant p53) and HCT116 colorectal cancer 29 cells (wild-type p53). Also, the authors demonstrated the effect of an extract in potentiating the induced cytotoxicity of DTX.

Please find below my comments/recommendation regarding your study.

All the name of chemicals/ acronym should be explained (i.e., DMSO NaHCO3, HEPES, …)

Figure 2: The authors demonstrated the viability levels of DU145, however they did not present the respective data for HCT116 under the same experimental conditions.

Figure 5: on the western immunoblots quantification I would recommend the authors to label the axis of the graphical representation. Also, statistical analysis should be performed between the various conditions.
Figure 6: I would recommend the authors to include the cell migration panels of flow cytometry rather the intensity panels. As it is more indicative of what actually happens to the cellular level. Also, graphs should be included with the percentages among with a statistical significance, between the various population.

I would recommend the authors to evaluate the ability of the extracts themselves or in combination with the used chemotherapeutic drug, to cause perturbation in the ROS levels. And if they do so, what is the correlation between the induced cell death.

A graphical abstract demonstrating the whole idea of this manuscript is highly recommended.

Author Response

"Please see he attachment."

Round 2

Reviewer 1 Report

Comments and Suggestions for Authors

I believe that the article can be published in its present form, the authors have made improvements to the article.

Reviewer 3 Report

Comments and Suggestions for Authors

I would like to thank the authors for taking into consideration my suggestions/advice.